# The Prognostic Value of Deleted in Colorectal Cancer (DCC) Receptor and Serum Netrin-1 in Severe Traumatic Brain Injury

**DOI:** 10.3390/jcm11133700

**Published:** 2022-06-27

**Authors:** Yuanda Zhang, Qiao Zhang, Lihua Sun, Dongxu Zhao, Cijie Ruan, Jue Zhou, Haoyuan Tan, Yinghui Bao

**Affiliations:** 1Department of Neurosurgery, Renji Hospital, School of Medicine, Shanghai Jiao Tong University, Shanghai 200120, China; zhangyuanda123@sjtu.edu.cn (Y.Z.); sunlihua6808@renji.com (L.S.); zhaodongxu@renji.com (D.Z.); ruan_cijie@sjtu.edu.cn (C.R.); j.zhou@sjtu.edu.cn (J.Z.); tanhy42.195@sjtu.edu.cn (H.T.); 2Shanghai Key Laboratory for Tumor Microenvironment and Inflammation, Department of Biochemistry and Molecular Cell Biology, School of Medicine, Shanghai Jiao Tong University, Shanghai 200025, China; zhangqiao@sjtu.edu.cn

**Keywords:** DCC receptor, Netrin-1, traumatic brain injury

## Abstract

Traumatic brain injury (TBI) is a common neurological disease. Netrin-1 and deleted in colorectal cancer (DCC) receptor are potential biomarkers associated with nerve regeneration and immune regulation. We aimed to investigate the ability of the DCC receptor and Netrin-1 to predict a high ICP level after operation in severe traumatic brain injury and their prognostic significance. This study is a prospective observational study. We selected 23 patients with traumatic brain injury who had undergone surgical operations as subjects. Immunohistochemical staining was performed on the contusion tissue that was removed by the operation to determine the expression of DCC receptor. At the same time, enzyme-linked immunosorbent assay (ELISA) kits were used to detect the serum Netrin-1 content. Determination of intracranial pressure (ICP) value was measured by intraventricular catheter. The Glasgow Outcome Scale (GOS) score at six months after trauma was defined as the main study endpoint. The results showed that serum Netrin-1 concentrations of patients in the critical TBI group (GCS 3–5 points) was significantly lower than that in the severe TBI group (GCS 6–8 points). The ICP peak and average mannitol consumption in the high Netrin-1 group were significantly lower than those in the low Netrin-1 group. DCC receptor-positive patients had a significantly lower ICP peak. There was no significant difference in six month-GOS scores between patients in the high and low Netrin-1 groups, while DCC receptor concentrations below 3.82 ng/mL predicted poor prognosis (GOS 1–3 points). In conclusion, the expression level of the DCC receptor can better evaluate the postoperative high ICP level and prognosis than the level of serum Netrin-1 in severe traumatic brain injury.

## 1. Introduction

Traumatic brain injury (TBI) is a common disease with high morbidity and has become a serious public health problem especially in developing countries. In China, the mean mortality rate of TBI is about 3 per 100,000 people [1]. Overall, age, Glasgow Coma Scale (GCS) score, pupillary light reflex, Computed Tomography (CT) findings, hypoxia and systemic hypotension are the significant factors determining the survival, although the mortality rate varies in different regions due to socioeconomic factors [2]. Traumatic brain injury includes two types: primary injury and secondary injury. Primary injury mainly refers to brain contusion, often associated with subarachnoid hemorrhage. Secondary injuries include posttraumatic brain edema and intracranial hematoma [3]. The causes of brain edema are very complex. In short, factors such as blood–brain barrier disruption, excitatory amino acid, calcium overload, hypoxia, and free radical generation play important roles [4]. Intracranial pressure (ICP) is the main monitoring indicator for the occurrence and development of brain edema after TBI. A large number of studies have shown that there is a close relationship between the degree of ICP and the TBI prognosis [5].

TBI is a disease with complex mechanisms and varying degrees. Therefore, finding effective monitoring methods and biomarkers is an important factor in the treatment of TBI. Highly sensitive biomarkers can not only advance the time window of treatment, but also provide an important reference for the selection of treatment methods. Classical biomarkers are often associated with cells or structural damage, such as neuronal damage markers neuron-specific enolase (NSE) [6], ubiquitin C-terminal hydrolase-L1 (UCH-L1) [7], astrocytes damage markers glial fibrillary acidic protein (GFAP) [8], S100B [9], axonal damage markers neurofilament protein (NF) [10] and myelin basic protein (MBP) [11]. Later studies found that certain autoantibodies [12] or neurodegeneration-related proteins [13] can also reflect the prognosis of TBI. However, there is no single biomarker with comprehensive and accurate predictive value, even showing opposite results in different cases. Therefore, a comprehensive analysis of changes to multiple biomarkers is a direction to evaluate the TBI prognosis.

Netrin-1, a kind of laminin-related protein closely related to nerve injury and repair, was originally discovered by Tessier-Lavigne in the study of dorsal neural tube development. The regulatory effect of Netrin-1 on nervous system development is reflected in the bidirectional induction of axons, and this effect depends on the differential expression of the deleted in colorectal cancer (DCC) receptor and Uncoordinated-5 (UNC5) receptor family. The UNC5B receptor is the most interesting receptor in the UNC5 receptor family. In growth cones expressing DCC receptors, Netrin-1 causes an attractive effect. Conversely, Netrin-1 induces a repulsive effect upon expression of the Uncoordinated-5B (UNC5B) receptors [14]. Subsequent studies found that the differential expression of Netrin-1 and its receptors played an important role in the regulation of the inflammatory process [15] and oxidative stress [16] in a variety of diseases. In a related study of TBI, it was observed that Netrin-1 levels in peripheral blood decreased after trauma, and this was associated with prognosis [17]. It is worth noting that some studies suggested that the decline of peripheral Netrin-1 levels was consumptive and did not reflect its true expression in disease [18]. Therefore, we need to more accurately evaluate the value of Netrin-1 in the prognostic assessment of TBI. In this study, we targeted the research goal to Netrin-1, the DCC receptor and the UNC5B receptor aiming at a deeper analysis of their roles in prognostic assessment and the objective evaluation of their potential to serve as biomarkers.

## 2. Materials and Methods

### 2.1. Patients

From June 2020 to March 2021, we collected 23 patients with traumatic brain injury (ICD-10 code: S06.902) who were admitted to Renji Hospital (Shanghai, China) for surgery. Inclusion criteria for the study included: severe traumatic brain injury (GCS 3–8), surgery performed within 6 h after trauma. Exclusion criteria for the study included: other diseases of the central nervous system, other severe systemic diseases (such as severe renal failure), postoperative severe central nervous system infection and loss-to-follow-up (inability to contact patients or their families).

In our study, contusion tissue removed during surgery was collected. Treatment strategies after TBI include: relieving cerebral edema to reduce intracranial pressure, preventing and treating hypotension and hypoxemia, improving cerebral perfusion, preventing infection, mild hypothermia therapy, sedation and analgesia and nutritional support. Among them, dehydration treatment is the main way to relieve cerebral edema. The hyperosmolar agents we used include: mannitol, glycerol fructose, hypertonic saline and furosemide. We chose a different combination of hyperosmolar agents for each patient according to the actual situation, although mannitol was the main component. For the convenience of statistics, we only counted the amount of mannitol to evaluate the sensitivity of patients to dehydration treatment. Changes in intracranial pressure were monitored by an intracranial probe placed in the lateral ventricle. In most cases, intracranial pressure reached a stable level by 7 days after surgery. Considering the risk of intraventricular infection, we removed intracranial pressure monitoring 10–14 days after surgery. In addition, all patients received 1–2 week sedation treatment (dexmedetomidine) according to the actual situation. This study was approved by the ethics committee of our hospital (RA-2020-214). A written informed consent was obtained from all the participants or their relatives.

### 2.2. Variables

The following variables were collected in this study: patient gender, age, underlying diseases (hypertension, diabetes or coronary heart disease), mean temperature of mild hypothermia therapy, average dose of dexmedetomidine, preoperative GCS score, preoperative Rotterdam computed tomography classification, the expression of DCC and UNC5B receptors in damaged tissues evaluated by immunohistochemistry, the protein content of DCC receptors in the damaged brain tissues, serum Netrin-1 level, postoperative ICP, mean daily dosage of mannitol during hospitalization and GOS score after 6 months. Adverse outcomes were defined as a GOS score of 1–3 at 6 months.

The underlying disease status of all patients was based on the medical history. Hypertension, diabetes and coronary heart disease were diagnosed and treated with medication by appropriate specialists. The diagnostic criteria for hypertension are: systolic blood pressure ≥ 140 mmHg or diastolic blood pressure ≥ 90 mmHg in the morning for three consecutive days in the absence of medication. The diagnostic criteria for diabetes are: fasting blood glucose ≥ 7.0 mmol/L, or two-hour postprandial blood glucose ≥ 11.1 mmol/L. The diagnostic criteria for coronary heart disease are: coronary angiography showing stenosis of more than 50%.

The specific items of Rotterdam computed tomography classification are shown in Table 1. The grading of postoperative ICP was determined based on the peak value obtained from postoperative ICP monitoring. The ICP peak was moderately elevated at 21–40 mmHg and severely elevated at >40 mmHg. The specific contents of the GOS score are as follows: 5 points (well recovered: return to normal life despite mild impairment); 4 points (mild disability: disabled but able to live independently; able to work under protection); 3 points (severe disability: remaining awake; disabled; needing attention in daily life); 2 (vegetative survival: only minimal response, such as eyes opening with cycles of sleep and wakefulness); 1 (death).

### 2.3. Immunohistochemistry

We took the intraoperatively removed brain tissues for immunohistochemical staining to determine the expression of DCC receptors and UNC5B receptors. Briefly, the tissues were fixed with 4% paraformaldehyde, then embedded in paraffin and sectioned, and then the sections were dewaxed, antigen retrieved and blocked with H_2_O_2_ blocking solution for 10 min. The sections were washed 3 times with PBS, and then incubated with 5% BSA for 10 min. After that, we added a primary antibody solution and incubated the sections overnight at 4 °C. After removing, the sections were washed 3 times with PBS, then added with the secondary antibody and incubated at room temperature for 1 h. After washing off the secondary antibody, the sections were added with chromogenic reagent and the chromogenic time was controlled under the microscope. Then, the sections were counterstained with hematoxylin and dehydrated with ethanol. The sections were observed and the pictures were taken under microscope.

### 2.4. ELISA Analysis

Determination of protein content of DCC receptors in the damaged brain tissues was performed using a commercially available ELISA kit (EH7828, FineTest, Wuhan, China). Briefly, we grounded the tissue into powder after treating with liquid nitrogen, then added 300 μL RIPA (Beyotime, Shanghai, China) and 10 μL PMSF (Beyotime, Shanghai, China) into 20 mg tissues, lysed on ice for 30 min, then took the supernatant after centrifugation and operated according to the kit instructions. Each sample was independently tested 3 times, and the final results were averaged.

Determination of serum Netrin-1 levels was performed using a commercially available ELISA kit (CSB-E11899h, CUSABIO, Wuhan, China). We collected peripheral venous blood from patients before surgery, centrifuged the supernatant, and then operated according to the kit instructions. Each blood sample was independently tested 3 times, and the final results were averaged.

### 2.5. Statistical Methods

SPSS 26.0 was used for statistical analysis. Numerical variables were summarized as mean ± SD (normal distribution) or median (nonnormal distribution). The values of categorical variables were presented as frequencies. Intergroup comparisons were performed using the Mann–Whitney U test or Fisher’s exact test where appropriate. Specifically, the Mann–Whitney U test was used for comparisons between two sets of numerical variables and Fisher’s exact test was used for comparisons between categorical variables. An operator characteristics (ROC) curve analysis was carried out to detect the predictive ability and to determine the cut-off value. Statistical significance was set at *p* < 0.05.

## 3. Results

### 3.1. Baseline Patient Characteristics

We enrolled a total of 23 patients with severe traumatic brain injury in the study. There were no significant differences in the subgroups in terms of age, gender, underlying diseases, mean temperature of mild hypothermia therapy and average dose of dexmedetomidine. Table 2 shows the baseline data of the patients.

### 3.2. The Relationship between Serum Netrin-1 Concentrations and Trauma Severity

Comparing serum Netrin-1 concentrations in patients with different Glasgow Coma Scale scores, our study showed that patients with lower GCS scores had lower serum Netrin-1 levels (Figure 1a). Comparing serum Netrin-1 concentrations in patients with severe vs. critical GCS score, we found that the serum Netrin-1 concentrations of patients in the critical TBI group (GCS 3–5 points) was significantly lower than that in the severe TBI group (GCS 6–8 points) (Figure 1b). In addition, we performed Rotterdam CT scores on patients based on imaging findings. Comparing serum Netrin-1 concentrations in patients with different Rotterdam CT scores, we found that patients with higher scores had lower serum Netrin-1 concentrations (Figure 1c).

### 3.3. Serum Netrin-1 Level Indicated Postoperative Intracranial Pressure and Sensitivity to Dehydration Therapy

Figure 2a showed the relationship between serum Netrin-1 concentrations and ICP peak among patients. Figure 2b showed the relationship between serum Netrin-1 concentrations and average mannitol consumption during hospitalization. According to the postoperative ICP peak, we divided the patients into moderately high ICP group (ICP 21–40 mmHg) and severe high ICP group (ICP > 40 mmHg). In the study, a total of 11 patients had severe ICP elevation (47.83%). Comparing serum Netrin-1 concentrations in patients with moderately high ICP vs. severe high ICP, we found that the serum Netrin-1 concentrations of patients in the severe high ICP group was significantly lower than that in the moderately high ICP group (Figure 2c). After grouping patients according to Netrin-1 levels and comparing ICP peak and average mannitol consumption in patients with high serum Netrin-1 level vs. low serum Netrin-1 level, we found that the ICP peak and average mannitol consumption in the high Netrin-1 group were significantly lower than those in the low Netrin-1 group (Figure 2d,e). Grouping both Netrin-1 levels and ICP peak simultaneously and comparing ICP level in patients with high Netrin-1 level vs. low Netrin-1 level, we found that 10% of patients in the high Netrin-1 group had severe ICP elevations (ICP > 40 mmHg) and 90% had moderate ICP elevations (ICP 21–40 mmHg). In the low Netrin-1 group, 76.92% of patients had severe ICP elevation (ICP > 40 mmHg) and 23.08% had moderate ICP elevation (ICP 21–40 mmHg) (Table 3).

### 3.4. Positive DCC Receptors Indicated Postoperative Intracranial Pressure and Sensitivity to Dehydration Therapy

To further explore the value of Netrin-1 in the assessment of TBI, we classified patients according to the differences in Netrin-1 receptor expression and compared ICP level in patients with high Netrin-1 level vs. low Netrin-1 level. In the high Netrin-1 group, 8 patients were DCC receptor-positive and 2 patients were DCC receptor-negative, 7 patients were UNC5B receptor-positive and 3 patients were UNC5B receptor negative. Among DCC receptor-positive patients, 100% had moderate ICP elevation. Among DCC receptor-negative patients, 50% had moderate ICP elevation and 50% had severe ICP elevation. Among UNC5B receptor-positive patients, 85.71% had moderate ICP elevation and 14.29% had severe ICP elevation. Among UNC5B receptor-negative patients, 100% had moderate ICP elevation. In the low Netrin-1 group, 2 patients were DCC receptor-positive and 11 were DCC receptor-negative, 8 patients were UNC5B receptor-positive and 5 patients were UNC5B receptor-negative. Among DCC receptor-positive patients, 100% had moderate ICP elevation. Among DCC receptor-negative patients, 9.1% had moderate ICP elevation and 90.91% had severe ICP elevation. Among UNC5B receptor-positive patients, 12.5% had moderate ICP elevation and 87.5% had severe ICP elevation. Among UNC5B receptor-negative patients, 40% had moderate ICP elevation and 60% had severe ICP elevation (Table 4).

Next, we grouped patients according to the expression of DCC receptors and UNC5B receptors and compared ICP peak and average mannitol consumption in patients with positive DCC expression vs. negative DCC expression and in patients with positive UNC5B expression and negative UNC5B expression. Compared with DCC receptor-negative patients, DCC receptor-positive patients had a significantly lower ICP peak and average mannitol consumption (Figure 3a,b). However, there was no significant difference between the UNC5B receptor-positive patients and the UNC5B receptor-negative patients in ICP peak and average mannitol consumption (Figure 3c,d). The rate of severe high ICP in the DCC receptor-positive group was 0%, while the rate of severe high ICP in the DCC receptor-negative group was 84.62%. The rate of severe high ICP in the UNC5B receptor-positive group was 46.67%, while the rate of severe high ICP in the UNC5B receptor-negative group was 50% (Table 5). The above results suggested that, compared with serum Netrin-1 levels, DCC receptors could better reflect the changes of postoperative intracranial pressure and the sensitivity of patients to mannitol treatment.

Considering the long time of immunohistochemical staining and the great influence of subjective factors of the examiner, we used the DCC receptor ELISA kit to evaluate the protein concentration of DCC receptors. Figure 4a showed the relationship between the protein concentration of DCC receptors and the postoperative intracranial pressure. Figure 4b showed the relationship between the protein concentration of DCC receptors and the average mannitol consumption. Next, we constructed ROC curves to assess the ability of DCC receptor concentrations and serum Netrin-1 concentrations to differentiate patients’ intracranial pressure levels. The results in Figure 4c showed that DCC receptor concentrations below 3.63 ng/mL predicted higher ICP levels with a sensitivity of 83%, a specificity of 96%, and a Youden J index of 0.79. Serum Netrin-1 concentrations below 0.52 ng/mL predicted higher ICP levels with a sensitivity of 83%, a specificity of 36% and a Youden J index of 0.19. Based on area under ROC curve (AUC), DCC receptor concentrations were better at predicting ICP levels than serum Netrin-1 concentrations.

### 3.5. Serum Netrin-1 Levels Did Not Evaluate Prognosis, While the Expression of DCC Receptors Reflected the Prognosis of TBI

Based on previous reports, we analyzed the relationship between serum Netrin-1 levels and 6 month-GOS scores. Comparing serum Netrin-1 concentrations in patients with GOS (4–5) vs. GOS (1–3), we found no significant difference in serum Netrin-1 concentrations between the poor prognosis group (GOS 1–3 points) and the good prognosis group (GOS 4–5 points) (Figure 5a). According to previous reports, we used 0.53 ng/mL as a standard to group serum Netrin-1 levels. Comparing prognosis evaluated by GOS scores in patients with high serum Netrin-1 level vs. low serum Netrin-1 level, we found that there was no significant difference in 6 month-GOS scores between patients in the high and low Netrin-1 groups (Table 6).

The results in Figure 5b showed the ability of DCC receptor concentrations and serum Netrin-1 concentrations to discriminate GOS levels. DCC receptor concentrations below 3.82 ng/mL predicted poor prognosis with a sensitivity of 93%, a specificity of 98% and a Youden J index of 0.91. Serum Netrin-1 concentrations below 0.33 ng/mL predicted poor prognosis with a sensitivity of 93%, a specificity of 50% and a Youden J index of 0.43. Based on the area under ROC curve (AUC), the ability of DCC receptor concentrations to predict prognosis was better than that of serum Netrin-1 concentrations.

## 4. Discussion

TBI is one of the most common traumatic diseases worldwide, and the main treatment is surgery. Since intracranial hypertension is one of the most important factors affecting prognosis, prevention and treatment of high ICP level is the most important part of various targeted treatment measures after surgery [5]. Among the current treatments for high ICP level, the most common treatment is dehydration. However, postoperative intracranial pressure changes rapidly, and it is difficult to grasp the timing of dehydration treatment. Therefore, intracranial pressure often develops to an uncontrollable level, which seriously affects the survival and recovery of patients. In this study, we evaluated the value of DCC receptors in predicting high ICP level and prognosis of TBI. Our study showed that the expression of DCC receptors was closely related to the ICP level, and patients with positive-DCC had a lower degree of postoperative high ICP level and were more sensitive to dehydration therapy. Further studies had shown that patients with positive-DCC had a better prognosis. Therefore, the DCC receptor may become a valuable TBI biomarker.

The most classical biological effect of the DCC receptor is to mediate axonal attraction and repulsion together with its ligand Netrin-1. This effect is dependent on the intracellular P3 domain of the DCC receptor. Netrin-1 mediates axonal attraction upon homodimerization of DCC receptors. In contrast, when DCC receptors heterodimerize with UNC5B receptors, Netrin-1 mediates axonal repulsion [19]. In addition, DCC receptors play an important role in maintaining the integrity of the blood–brain barrier. In a study of subarachnoid hemorrhage, Zongyi Xie et al. found that Netrin-1 upregulated the expression of cell junction-associated proteins through DCC receptors, thereby preventing the breakdown of the blood–brain barrier, and DCC siRNA treatment could counteract the effects of Netrin-1 [20]. Our study observed a close relationship between DCC receptors and postoperative high ICP level in clinical cases. The destruction of the blood–brain barrier is one of the most important factors in the occurrence and development of postoperative cerebral edema [4]. The protective effect of DCC receptors on the blood–brain barrier is one possible explanation for our findings. Despite the lack of relevant basic studies in animal models of TBI, our results still suggest that the DCC receptor is a valuable target in therapeutic strategies for high ICP level.

The clinical predictive value of Netrin-1 has been validated in a variety of CNS diseases. In an ischemic stroke, Yuhan Zang et al. found that high serum Netrin-1 levels could reduce the risk of stroke and were associated with better prognosis [21]. This is similar to the conclusion obtained in TBI17. Notably, in both ischemic stroke [22] and subarachnoid hemorrhage [23] models, up-regulation of Netrin-1 and its receptors was observed around the injury site, which was opposite to the changes in peripheral blood. A possible explanation is that Netrin-1 is involved in the inflammatory response process and inhibits the progression of inflammation, which leads to the depletion of peripheral blood Netrin-1 [15]. Such results reflect the complex role of Netrin-1 in a variety of pathophysiological processes. Although our study confirmed the above conclusions, it also found that the predictive value of serum Netrin-1 was not as strong as brain tissue-derived DCC receptors. Considering that most TBI patients do not require surgery, serum Netrin-1 is a convenient and effective biomarker. However, for surgical patients, we recommend the expression of DCC receptors in brain tissue as part of prognostic assessment.

Although no studies have reported the value of DCC receptors in the prognostic assessment of TBI, a large number of basic studies have found the protective role of DCC receptors in a variety of CNS injury [24]. The most prominent role of DCC receptors is in the regulation of apoptosis. In stroke-related studies, it has been reported that DCC receptors attenuate hypoxia-induced apoptosis and DNA damage [25]. Interestingly, DCC receptors can induce apoptosis in the absence of Netrin-1, whose mechanism is currently unknown [26]. Therefore, the upregulation of Netrin-1 and DCC receptors may be an endogenous protective effect after cerebral ischemia or hemorrhage. Similar conclusions were also confirmed by our study. In the high Netrin-1 group, the positive rate of DCC receptors was also increased, which reflected the synergistic effect between Netrin-1 and DCC receptors. Another important role of DCC receptors is to promote the regeneration of blood vessels and nerves around the site of injury [27]. In the middle cerebral artery occlusion model, studies have found that the axonal network shaped by DCC receptors is beneficial to angiogenesis [28]. This role reflects the important value of DCC receptors in the process of injury recovery. Our study confirmed a positive relationship between the presence of DCC receptors and GOS scores. The mechanism needs further basic research to confirm.

It is worth noting that our study was a single-center study and lacked representation of other practice settings. In addition, our sample size was small and no longer follow-up was performed. Following studies should include a larger sample size and conduct multiple comparisons, so as to obtain more general conclusions.

## 5. Conclusions

Our study investigated samples from patients’ peripheral blood and brain tissues around the site of injury. Our analysis showed that high expression of DCC receptors in injured tissues was closely associated with a lower degree of postoperative high ICP level. At the same time, the positive rate of DCC receptors is an indicator of TBI prognostic evaluation. A high positive rate of DCC receptors suggested a better prognosis. Compared with serum Netrin-1 levels, DCC receptors have a more accurate prognostic value, especially for TBI patients who require surgery. Taken together, our study provides a possible biomarker for the selection of treatment options and prognostic assessment after TBI. Despite our small sample size, the high accuracy of DCC receptors is an attractive research direction.

## Figures and Tables

**Figure 1 jcm-11-03700-f001:**
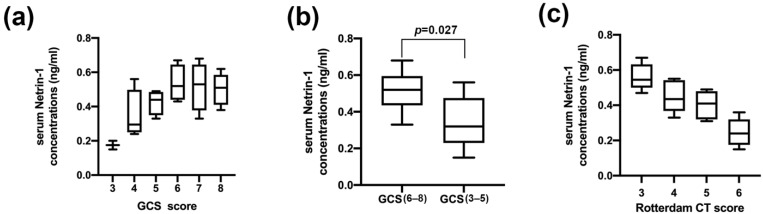
The relationship between serum Netrin-1 concentrations and trauma severity. (**a**) Comparisons of serum Netrin-1 concentrations in patients with different Glasgow Coma Scale scores. (**b**) Differences in terms of serum Netrin-1 concentrations between critical TBI group (GCS 3–5 points) and severe TBI group (GCS 6–8 points). (**c**) Comparisons of serum Netrin-1 concentrations across Rotterdam computed tomography classification.

**Figure 2 jcm-11-03700-f002:**
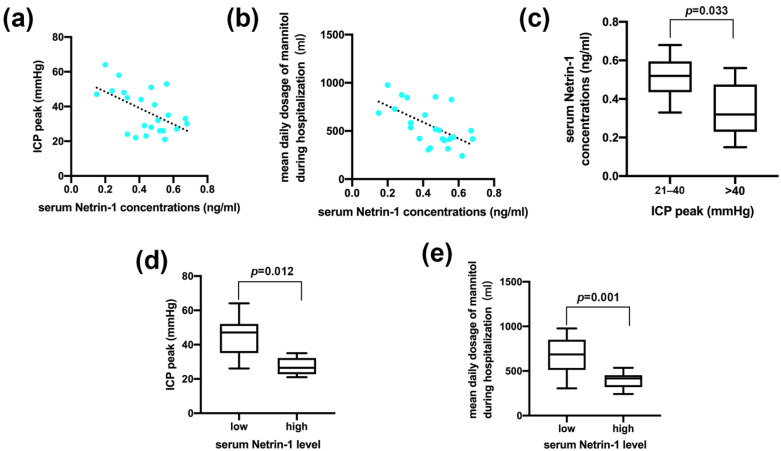
Serum Netrin-1 level indicated postoperative intracranial pressure and sensitivity to dehydration therapy. (**a**) The relationship between serum Netrin-1 concentrations and ICP peak among patients. (**b**) The relationship between serum Netrin-1 concentrations and average mannitol consumption during hospitalization. (**c**) Differences in terms of serum Netrin-1 concentrations between moderately high ICP group (ICP 21–40 mmHg) and severe high ICP group (ICP > 40 mmHg). (**d**) Differences in ICP peak between the high Netrin-1 group and the low Netrin-1 group. (**e**) Differences in average mannitol consumption between the high Netrin-1 group and the low Netrin-1 group.

**Figure 3 jcm-11-03700-f003:**
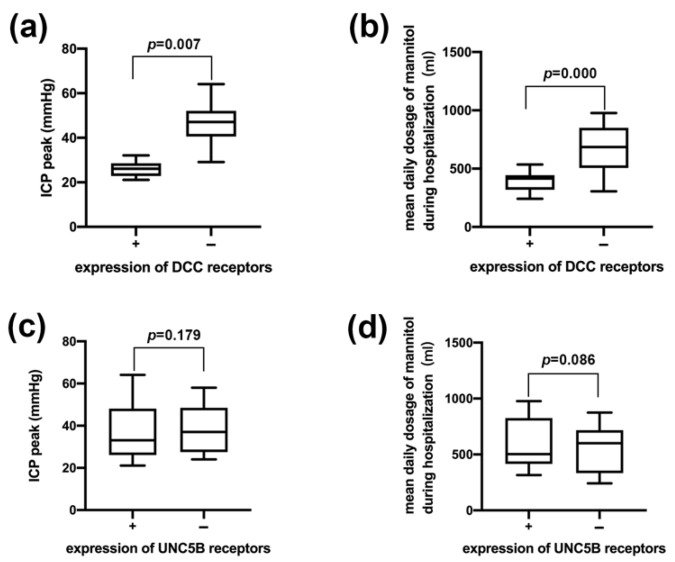
Comparisons of the postoperative intracranial pressure and sensitivity to dehydration therapy in patients with different expression of DCC receptors and UNC5B receptors. (**a**) Comparison of the postoperative intracranial pressure in patients with different expression of DCC receptors. (**b**) Comparison of the average mannitol consumption in patients with different expression of DCC receptors. (**c**) Comparison of the postoperative intracranial pressure in patients with different expression of UNC5B receptors. (**d**) Comparison of the average mannitol consumption in patients with different expression of UNC5B receptors.

**Figure 4 jcm-11-03700-f004:**
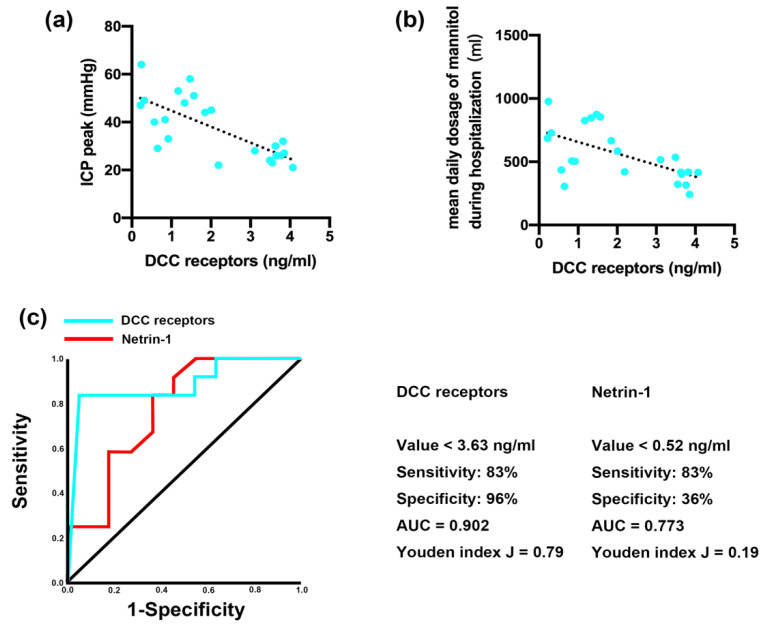
The respective relations between the protein concentration of DCC receptors and the postoperative intracranial pressure; sensitivity to dehydration therapy. (**a**) The relationship between the protein concentration of DCC receptors and the postoperative intracranial pressure. (**b**) The relationship between the protein concentration of DCC receptors and the average mannitol consumption. (**c**) Receiver operating characteristic curve for analyzing predictive ability regarding protein concentration of DCC receptors and serum Netrin-1 concentrations for high postoperative intracranial pressure.

**Figure 5 jcm-11-03700-f005:**
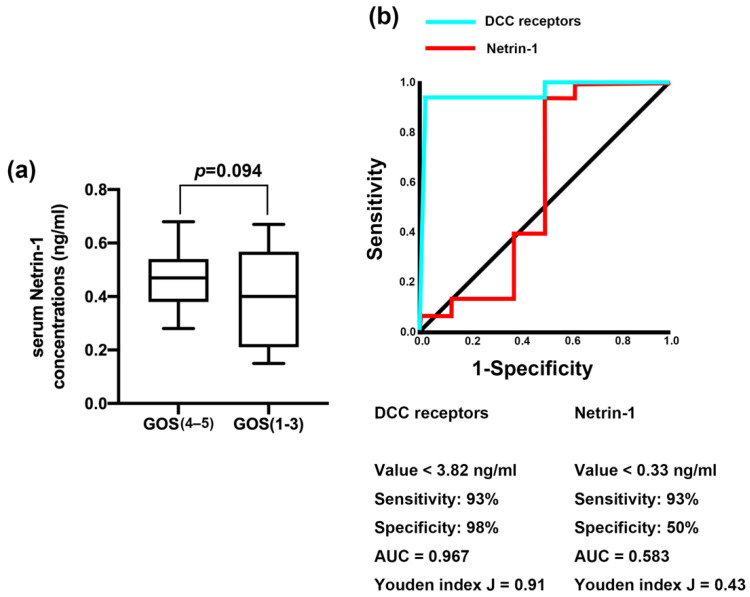
Serum Netrin-1 levels did not evaluate prognosis, while the expression of DCC receptors reflected the prognosis of TBI. (**a**) Differences in terms of serum Netrin-1 concentrations between poor prognosis group (GOS 1–3 points) and good prognosis group (GOS 4–5 points). (**b**) Receiver operating characteristic curve for analyzing predictive ability regarding protein concentration of DCC receptors and serum Netrin-1 concentrations for poor prognoses.

**Table 1 jcm-11-03700-t001:** Rotterdam computed tomography classification.

CT Findings	Score
Basal cisterns	
Normal	0
Compressed	1
Absent	2
Midline shift of the brain parenchyma	
≤5 mm	0
≥5 mm	1
Epidural hematoma, brain contusion or other occupying lesions	
Yes	0
No	1
Intraventricular hemorrhage or subarachnoid hemorrhage	
Yes	0
No	1
Total	+1

CT = computed tomography.

**Table 2 jcm-11-03700-t002:** Baseline Patient Characteristics.

	All Patients (*n* = 23)	ICP Level		GOS Score	
		Moderate	Severe	Unfavorable	Favorable
		(*n* = 13)	(*n* = 10)	(*n* = 15)	(*n* = 8)
Gender (male/female)	11/12	6/7	5/5	7/8	4/4
Age (y)	49.48 ± 15.65	44.15 ± 12.79	56.40 ± 16.30	44.87 ± 13.38	58.13 ± 15.92
Hypertension	11 (47.83%)	4 (30.77%)	7 (70.00%)	7 (46.67%)	4 (50.00%)
Diabetes mellitus	8 (34.78%)	5 (38.46%)	3 (30.00%)	6 (40.00%)	2 (25.00%)
Coronary heart disease	4 (17.39%)	2 (15.38%)	2 (20.00%)	2 (13.33%)	2 (25.00%)
Mean temperature of mild hypothermia therapy (°C)	33.54 ± 0.44	33.44 ± 0.39	33.65 ± 0.46	33.59 ± 0.54	33.52 ± 0.37
Average dose of dexmedetomidine (μg/kg/d)	15.18 ± 1.87	15.12 ± 1.94	15.24 ± 1.78	14.83 ± 1.77	15.36 ± 1.89

ICP = intracranial pressure; GOS = Glasgow Outcome Scale.

**Table 3 jcm-11-03700-t003:** Comparison of ICP level between the high Netrin-1 group and the low Netrin-1 group.

	ICP Level (mmHg)		Total	
	21–40	>40		
Low Netrin-1	3	10	13	*p* = 0.001
High Netrin-1	9	1	10	
Total	12	11	23	

ICP = intracranial pressure.

**Table 4 jcm-11-03700-t004:** Comparisons of ICP level between the high Netrin-1 group and the low Netrin-1 group (considering the expression of DCC and UNC5B receptors).

		ICP Level (mmHg)		Total	
		21–40	>40		
High Netrin-1 (*n* = 10)	DCC (+)	8	0	8	*p* = 0.035
	DCC (−)	1	1	2	
	Total	9	1	10	
	UNC5B (+)	6	1	7	*p* = 0.490
	UNC5B (−)	3	0	3	
	Total	9	1	10	
Low Netrin-1 (*n* = 13)	DCC (+)	2	0	2	*p* = 0.005
	DCC (−)	1	10	11	
	Total	3	10	13	
	UNC5B (+)	1	7	8	*p* = 0.252
	UNC5B (−)	2	3	5	
	Total	3	10	13	

ICP = intracranial pressure; DCC = deleted in colorectal cancer; UNC5B = Uncoordinated-5B.

**Table 5 jcm-11-03700-t005:** Comparison of ICP level between the DCC-positive group and the DCC-negative group. Comparison of ICP level between the UNC5B-positive group and the UNC5B-negative group.

	ICP Level (mmHg)		Total	
	21–40	>40		
DCC (+)	10	0	10	*p* = 0.000
DCC (−)	2	11	13	
Total	12	11	23	
UNC5B (+)	8	7	15	*p* = 0.879
UNC5B (−)	4	4	8	
Total	12	11	23	

ICP = intracranial pressure; DCC = deleted in colorectal cancer; UNC5B = Uncoordinated-5B.

**Table 6 jcm-11-03700-t006:** Comparison of GOS level between the high Netrin-1 group and the low Netrin-1 group.

	GOS Score		Total	
	1–3	4–5		
Low Netrin-1	7	6	13	*p* = 0.192
High Netrin-1	8	2	10	
Total	15	8	23	

GOS = Glasgow Outcome Scale.

## Data Availability

Data can be accessed by contacting the corresponding author on reasonable request.

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
