# Peer review of "The Prognostic Value of Deleted in Colorectal Cancer (DCC) Receptor and Serum Netrin-1 in Severe Traumatic Brain Injury"

_jcm, 2022, doi:10.3390/jcm11133700_

Round 1

Reviewer 1 Report

I appreciate the authors presenting this research article emphasizing the important role of DCC and Netrin-1 on TBI. My comments are as follows

1. The major drawback is the small cases number. 

The methods must be improved as follows

1. What were the types of brain haemorrhage or injury after traumatic brain injury?

2. Were sedations used in this study?

3. What is the guideline of post-operative treatment?

3. How many days of post-operative ICP monitoring?

4. Please provide the peak intracranial pressure per day for the first three     days in relation to DCC for each day.

5. Please provide the peak intracranial pressure per day for the first three days in relation to Netrin-1 for each day.

6. The limitation of current study

Reviewer 2 Report

The present article explored the prognostic value of DCC 2 Receptor and Serum Netrin-1 in severe traumatic brain injury. The findings of this study might enhance the armamentarium of prognostic biomarkers in severe TBI.

Please, define every abbreviation at 1st mention both in the abstract and in the main text (e.g., GOS score, GCS). Also, each Table should stand independently (define every abbreviation in the footnotes.)

2.1. Patients: Participant selection: How were participants enrolled (icd-10 coding? Manual search?)? What were the reasons for loss-to-follow up?

2.2. Variables: Please define the documented variables (hypertension, diabetes or coronary heart disease, Rotterdam computed tomography classification, postoperative ICP categorization to moderate and severe, GOS score and its categorization). Definitions should be given in the methods and not the results section.

Moreover, how were the aforementioned variables handled? You should describe your analyses before performing them, e.g., comparing Netrin-1 levels in patients with critical vs. severe GCS score, correlating Netrin-1 levels with Rotterdam CT scores, etc. Also, provide a rationale for the categorization of scale variables?

Was there an association between Netrin-1 and presence of DCC receptors?

UNC5B receptors are first mentioned in the methods section. You should add a relevant description in the introduction (and explain your choice of measuring these 3 laboratory indices – their connection).

Provide the statistical test used for correlation analyses.

Describe the limitations of your study (study sample, multiple analyses, extrapolation-generalizability considerations).

Round 2

Reviewer 1 Report

The revised manuscript has corrected according to my comments point-by point. Accept is my final decision. 

Author Response

Thank you very much for your praise and comments!

Reviewer 2 Report

Thank you for considering my suggestions. Please replace the word correlation throughout the text. Since your statistical approach did not include correlation statistics, it would be better to avoid this term (it denotes linear association).

Author Response

We sincerely thank you for examining the manuscript and providing constructive comments. We have replaced the word "correlation" in the revised manuscript based on your suggestion. Any changes in our revision have been highlighted using a red-colored text in the revised manuscript. Specifically, the modified text was located in "Abstract", "Results" and the figure legends of Figure 1, Figure 2, Figure 3, and Figure 4.